# Dynamical trade-offs arise from antagonistic coevolution and decrease intraspecific diversity

Weini Huang[1,2,3], Arne Traulsen [2], Benjamin Werner[2,4], Teppo Hiltunen[5] & Lutz Becks [6]

Trade-offs play an important role in evolution. Without trade-offs, evolution would maximize fitness of all traits leading to a "master of all traits". The shape of trade-offs has been shown to determine evolutionary trajectories and is often assumed to be static and independent of the actual evolutionary process. Here we propose that coevolution leads to a dynamical trade-off. We test this hypothesis in a microbial predator–prey system and show that the bacterial growth-defense trade-off changes from concave to convex, i.e., defense is effective and cheap initially, but gets costly when predators coevolve. We further explore the impact of such dynamical trade-offs by a novel mathematical model incorporating de novo mutations for both species. Predator and prey populations diversify rapidly leading to higher prey diversity when the trade-off is concave (cheap). Coevolution results in more convex (costly) trade-offs and lower prey diversity compared to the scenario where only the prey evolves.

---

[1] Group of Theoretical Biology, The State Key Laboratory of Biocontrol, School of Life Science, Sun Yat-sen University, Guangzhou 510060, China. [2] Department of Evolutionary Theory, Max Planck Institute for Evolutionary Biology, August-Thienemann-Straße 2, 24306 Plön, Germany. [3] Evolution and Cancer Lab, Tumour Biology, Barts Cancer Institute, Queen Mary University of London, Charterhouse Square, London EC1M 6BQ, UK. [4] Centre for Evolution and Cancer, The Institute of Cancer Research, 5 Cotswold Rd Sutton, London SM2 5NG, UK. [5] Department of Microbiology, University of Helsinki, P. O. Box 5600014 Helsinki, Finland. [6] Community Dynamics Group, Department of Evolutionary Ecology, Max Planck Institute for Evolutionary Biology, August-Thienemann-Straße 2, 24306 Plön, Germany. Correspondence and requests for materials should be addressed to W.H. (email: weini.huang@qmul.ac.uk)

Trade-offs exist when there are negative correlations between different traits of an organism. They play an important role in maintaining polymorphisms in populations, where individuals invest in traits differently[1]. One critical trade-off for the evolution of species is between survival and reproduction, e.g., the growth-defense trade-off of a prey population, the growth-resistance trade-off of a host population, or the growth-offense trade-off of a predator or parasite population[2].

A growing number of studies show that evolutionary and ecological changes can occur on similar time scales[3–8] and researchers have moved beyond classical ecological models assuming static traits of species over ecological time scales[9–13]. Rapid prey evolution can, for example, shift predator-prey cycles from the classical one-quarter phase lag towards longer cycles, which are nearly out of phase[14–16]. Trade-offs and the shape of the growth-defense trade-off curve among different prey types can further determine the oscillatory behaviors of predator and prey densities over time;[15, 17–21] different efficiencies of prey defense and associated fitness costs can lead to cryptic cycles, where the predator abundance changes, but the total prey abundance remains constant[22], steady state dynamics where cycling stops[23], or even reversals in the direction of the predator-prey cycles[20]. Trade-offs hence not only play a major role for the maintenance of polymorphisms within prey populations but may also change their ecological dynamics over time.

Previous studies have shown the importance of trade-offs with rapid evolution within a single species, for example, in the evolution of only the prey[15, 17–21, 24]. Here we focus on the impact of trade-offs in coevolving systems. Coevolution is likely the norm in natural environments where species sharing the same environment will encounter and consequently interact with each other[12]. Coevolution might change the evolutionary potential of species over time[25, 26], as an adaptation in one species might result in significant fitness gains initially, which may decrease as its opponent species counter-adapts after some time. We hypothesize that the shapes of trade-off curves are not static as assumed in previous studies, but depend on the evolutionary potential of both species and can become dynamical with coevolution. The dynamical shape of the trade-off is then predicted to further affect the population dynamics of the coevolving populations, e.g., the pattern of predator-prey cycles[15, 16]. Specifically, multiple prey types coexist and out-of-phase cycles are more prevalent between the total prey and predator abundances when the trade-off curve is more concave and defense is cheap.

We first test this dynamical trade-off hypothesis in a microbial predator-prey system after bacteria and ciliate populations (*Pseudomonas fluorescens* and *Tetrahymena thermophila*) were allowed to coevolve for many generations. Our experimental results confirm that the growth-defense trade-off within the bacteria populations changed depending on the coevolutionary history. Defense was efficient and cheap in the presence of naive ciliates, but less efficient and costly in the presence of coevolved ciliates. Coevolution led to a shift from a cheap to a costly trade-off. Such a dynamical trade-off can potentially be important for the evolutionary and the ecological dynamics of species. Thus, we further investigated the consequences of a dynamical trade-off systematically using a mathematical model. We incorporated de novo mutations as stochastic events in prey and predator populations[27]. This allowed us to describe the coevolutionary process without imposing a predefined number of prey or predator types. We also considered feedbacks between the evolutionary process and demographic fluctuations[28], where the carrying capacities are not fixed parameters but change with the coevolution of the prey and predator types. Consistent with our experiment, starting from naive populations, the trade-off curve changes from an initial shape of relatively cheap defense to be more costly with coevolution. This ultimately leads to a lower prey diversity compared to only prey evolution. Interestingly, the predator diversity does not monotonically increase with prey diversity due to the dynamical trade-off. Together, our experimental and theoretical results show that dynamical trade-offs arise and significantly affect the evolution and maintenance of intraspecific diversity.

## Results

**Bacteria-ciliate coevolution leads to a dynamical trade-off.** To test our hypothesis that coevolution leads to a dynamical trade-off, i.e., a shift in the shape of the trade-off curve, we measured growth rates and defense against consumption by predatory ciliates for different bacteria and ciliate populations isolated from long-term coevolution selection lines. Replicated selection lines were started from the same ancestral isogenic ciliate stock and were allowed to coevolve together with bacteria populations which were started from the same isogenic stock (hereafter: coevolved ciliates) or were kept as naive ciliates in a bacteria-free environment (hereafter: ancestral ciliates). We isolated coevolved ciliate populations from two selection lines after ~550 generations. At the same time, we isolated bacteria populations (after ~1000 generations) from the same and one additional coevolved selection line. We then measured bacterial growth and defense of the coevolved and ancestral bacteria populations, when exposed to the ancestral or the two coevolved ciliate populations in a factorial design (see Methods section). We used ciliate growth rates as proxy for bacterial defense as in previous studies[14, 21, 22].

We found that bacterial growth rates were significantly higher in the ancestral bacteria than the coevolved bacteria (Kruskal–Wallis ancestral vs. coevolved bacteria: $\chi^2 = 19.8$, df = 1, $p = 1.275 \times 10^{-7}$). Defense against the naive ciliates was significantly lower for ancestral compared to coevolved bacteria (Fig. 1a, Kruskal–Wallis ancestral vs. coevolved bacteria: $\chi^2 = 6.2$, df = 1, $p = 0.005$). Furthermore, coevolved ciliates had a significantly higher growth rate compared to the ancestral ciliates at a given coevolved bacteria population (Fig. 1b–d, coevolved bacteria population 1: Kruskal–Wallis ancestral vs. coevolved ciliate: $\chi^2 = 5.4$, df = 1, $p = 0.007$; bacteria population 2: Kruskal–Wallis ancestral vs. coevolved ciliate: $\chi^2 = 5.4$, df = 1, $p = 0.007$; bacteria population 3: Kruskal–Wallis ancestral vs. coevolved ciliate: $\chi^2 = 4.3$, df = 1, $p = 0.025$).

We inferred the trade-off curves by applying a power function ($\propto z^c$) to a non-linear regression between bacteria and ciliate growth. We used the bacteria growth as the base $z$, and the mean values of ciliate growth in three technical replicates (the same symbol and the same color in Fig. 2) as the power (outcome) of the function. The estimated exponent $c$ indicates the trade-off shape, with $c > 1$ as a concave (cheap) shape and $c < 1$ as a convex (costly) shape. Examining the trade-off among different bacteria populations in the presence of naive ciliates, we observed the expected concave shape (cheap trade-off, $c = 4.296$, $R^2 = 0.999$, Fig. 2 black curve) where a small decrease in prey growth severely improves defense and thus decreases the growth of ancestral ciliates. Interestingly, while the measurements in the two coevolved ciliate populations were similar (Kolmogorov–Smirnov-test, $p = 0.16$), they were significantly different from those in the naive ciliate population (Kolmogorov–Smirnov-test, $p = 0.02$). In coevolved ciliate populations, a similar cost in bacteria growth had an almost negligible effect on improving bacteria defense and the shape of the trade-off became convex (Fig. 2, green $c = 0.233$, $R^2 = 0.997$ and blue curve $c = 0.141$, $R^2 = 0.998$). Our results suggest that the predator-prey coevolution can indeed result in a dynamical trade-off curve

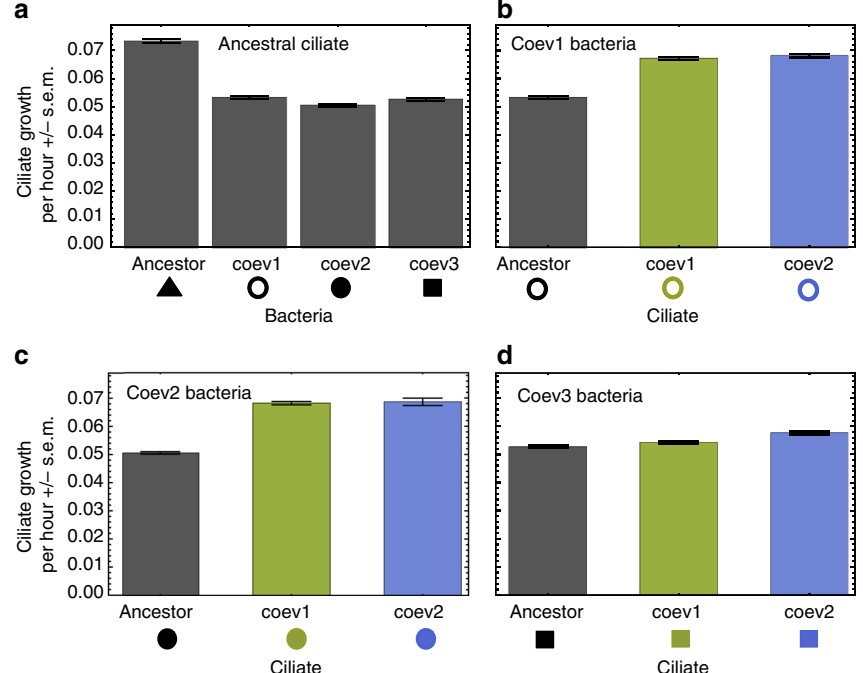

**Fig. 1** Increase in bacteria defense and ciliate predation ability in coevolving predator-prey system. **a** Mean growth rate of the ancestral ciliate population (*Tetrahymena thermophila*) is higher when growing on ancestral than coevolved bacteria populations (*Pseudomonas fluorescence*; isolated after ~1000 bacteria generations of co-culturing with the ciliate, Kruskal–Wallis test, $\chi^2 = 19.8$, df = 1, $p = 1.275 \times 10^{-7}$). When growing on three coevolved bacteria populations, mean ciliate growth rates of the ancestral ciliate populations are lower than the growth rates of two coevolved ciliate populations (isolated after ~550 ciliate generations of co-culturing with the bacteria), **b** coevolved bacteria population one marked by empty circles (Kruskal–Wallis test, $\chi^2 = 5.4$, df = 1, $p = 0.007$), **c** coevolved bacteria population two marked by filled circles (Kruskal–Wallis test, $\chi^2 = 5.4$, df = 1, $p = 0.007$), **d** coevolved bacteria population three marked by filled squares (Kruskal–Wallis test, $\chi^2 = 4.3$, df = 1, $p = 0.025$). Shown are mean and standard errors of 3 technical replicates. Symbols below the *x*-axes correspond to the data points in Fig. 2

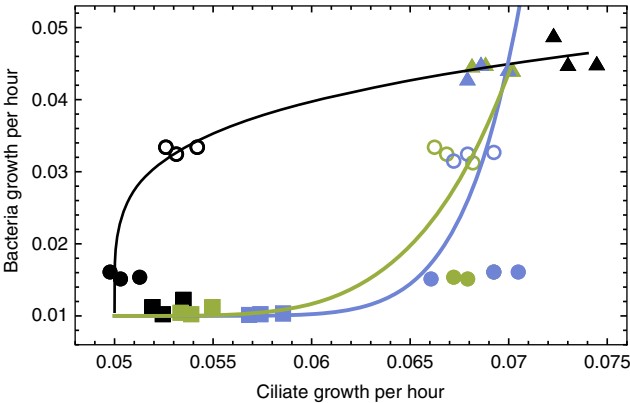

**Fig. 2** Growth-defense trade-off curves evolve from concave to convex in coevolving microbial predator–prey systems. Colors are associated to different ciliate populations and symbols to different bacteria populations. The growth and defense of four bacteria populations were measured in the presence of ancestral naive ciliates (black) and two independently coevolved ciliate populations (green coevo1 and blue coevo2). Bacteria defense is shown as cilliate growth rate in the x-axis, where lower ciliate growth rates indicate higher bacteria defense. The triangles represent the ancestor bacteria population, which has the highest growth. The three coevolved bacteria populations are respectively shown by hollow circles, filled circles and squares, and they differ in growth (their values in y-axis). In the ancestral ciliates, a concave trade-off curve was selected (non-linear least squares fitting, black curve, $R^2 = 0.999$), and in coevolved ciliate populations convex trade-off curves were selected (nonlinear least squares fitting, green curve $R^2 = 0.997$ and blue curve $R^2 = 0.998$)

with a shift from a more cheap (ancestral predators) to a more costly shape (coevolved predators).

**Modeling the impact of dynamical trade-offs.** To further explore the impact of a dynamical trade-off curve on predator-prey coevolution, we modeled coevolution incorporating random de novo mutations and demographic fluctuations. Instead of defining a fixed number of prey and predator types, we started with homogenous populations of a single prey and predator type. New prey and predator types arise from mutations, which can spread, persist or go extinct due to selection and random drift. All possible events in predator and prey populations, birth (including mutation), death, competition, and predation, occur stochastically with certain rates over time (see details in Methods section and Supplementary Note 1 for a discussion of parameter choice).

The growth-defense trade-off for prey types in our model emerges from two types of microscopic interactions. The first one,

$$X_i \xrightarrow{b_x g_i} X_i + X_i, \tag{1}$$

describes the growth of a prey type $X_i$ with a type-specific rate $g_i \in (0, 1]$ scaled by a background birth rate $b_x$, which is the same for all prey types. As the prey growth rate is not infinitely large, its upper boundary can be any arbitrary finite positive number. Here we scale $g$ values between 0 and 1. When a prey reproduces, a mutation may occur with a small probability $\mu_x$. The mutant is characterized by a new $g$ value drawn randomly from a uniform distribution between 0 and 1. Alternatively, we also considered mutations that are derived from a normal distribution around the parental type (see Supplementary Note 2).

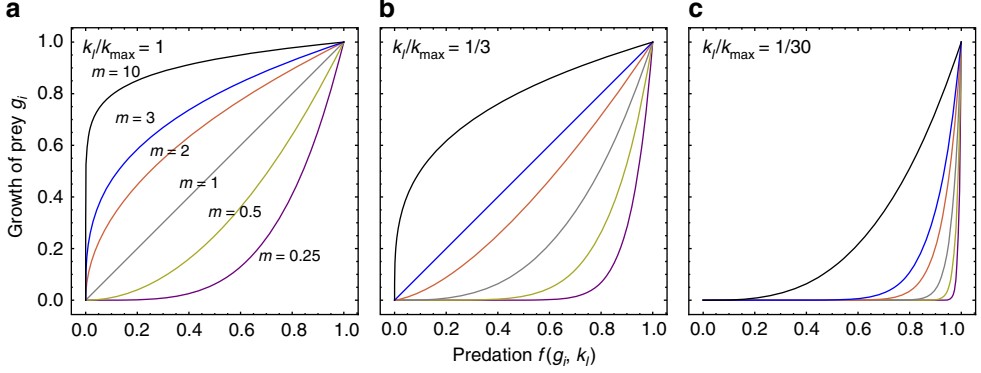

**Fig. 3** The trade-off between predation $f$ and prey growth $g$. **a** Illustration of the trade-off function between defense and prey growth in a naive predator population with the highest reproduction efficacy $k_{max}$. When $m = 1$, the trade-off function is linear, i.e., an increase in the growth will decrease the defense capability linearly and thus increase the predation rate linearly. When $m > 1$, the trade-off function is concave and defense is cheap. When $m < 1$, it is convex and defense is costly. The color refers to different $m$ values marked beside each curve in **a**. If the predator species does not evolve, the trade-off curves will remain the same as in this panel. If the predator coevolves with the prey, the trade-off curves change, e.g., **b**, **c** show the trade-off curves under evolved predators with higher predation abilities, i.e., $k_l/k_{max}$ equals to 1/3 or 1/30, respectively. With coevolution, the shapes of the trade-off functions will change continuously in different directions, depending on the present predator types

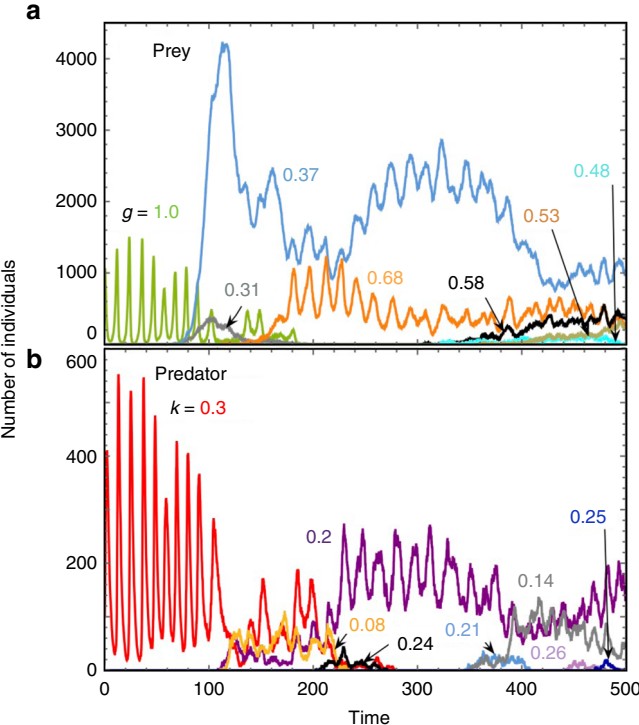

**Fig. 4** Stochastic coevolutionary dynamics of the prey and predator species. Different prey (**a**) and predator (**b**) types coexist with each other. The ancestor prey ($g_1 = g_{max} = 1.0$) and the ancestor predator ($k_1 = k_{max} = 0.3$) follow classical predator-prey cycles. New prey and predator types arise by de novo mutations. Some prey and predators invade and are maintained until they ultimately are lost again. Their $g$ and $k$ values are marked beside the corresponding types (parameters $b_x = 1.0$, $d_x = 0.1$, $\mu_x = 0.0001$, $r_c = 0.00005$, $X_1(0) = 1000$, $Y_1(0) = 100$, $d_y = 0.5$, $\mu_y = 0.001$, $p = 0.005$, $m = 3$)

The prey defense is captured in a second set of interactions, which also define the predation of a predator type $Y_l$ on a prey type $X_i$

$$X_i + Y_l \xrightarrow{k_l f(g_i, k_l)} Y_l + Y_l \qquad (2a)$$

and

$$X_i + Y_l \xrightarrow{(1-k_l)f(g_i, k_l)} Y_l. \qquad (2b)$$

Here, $f(g_i, k_l)$ is the predation rate, and $k_l$ is the ratio of predator growth to predation and represents the reproduction efficacy of a predator type $Y_l$. As a predator needs to consume to reproduce, we have $k_l \in (0, k_{max}]$ and $k_{max} < 1$. If $k_{max} = 1$, a predator reproduces every time it consumes. Here we use small $k_{max}$ and discuss the impact of larger $k_{max}$ in Supplementary Note 3. With a probability $\mu_y$, a predator produces a mutant with a new $k$ value drawn from a uniform distribution between 0 and $k_{max}$ (see Supplementary Note 2 for an extension to normal distributed mutations around parent types).

The predation rate of predator $Y_l$ on prey $X_i$, $f(g_i, k_l)$, depends on both the prey's defense level and the predator's predation capability. We assume there are energy constrains for both prey and predator. A faster growing prey has a lower defense and therefore enables higher predation rates on itself. If a predator has a higher reproduction efficacy, its predation rate decreases. These trade-offs imply that $f(g_i, k_l)$ is an increasing function of $g_i$ and a decreasing function of $k_l$. One simple choice that satisfies both requirements is $f(g_i, k_l) = p g_i^{m \frac{k_l}{k_{max}}}$, where $p$ is a constant that defines the time scale of predation, and $m$ determines the initial shape of the growth-defense trade-off (Fig. 3). Note, we used a similar power function to fit the trade-off curves from our experimental data and found a good agreement between the function and the data. In addition, while we have measured the growth-defense trade-off in prey in our experiments, the trade-off in predators is currently an assumption in our model.

To remain consistent with previous work[16, 29], we plot the defense (the predation function), $f(g_i, k_l)$, in the $x$-axis and the growth, $g_i$, in the $y$-axis, thus concave (cheap) and convex (costly) trade-offs refer to the shape of the trade-off curve in Fig. 3 instead of the predation function (see Methods section). When the growth-defense trade-off is concave, a small cost of prey growth leads to a large gain of defense, thus we say the defense is cheap; when it is convex, defense is costly. Under a naive predator $k_l = k_{max}$, $m < 1$ refers to a convex curve, $m = 1$ to a linear curve, and $m > 1$ to a concave curve.

Here we focus on dynamical trade-offs caused by coevolution and the shape of the predation function is fixed for a given $k_l$ (predator) in our model. While possible mechanisms other than

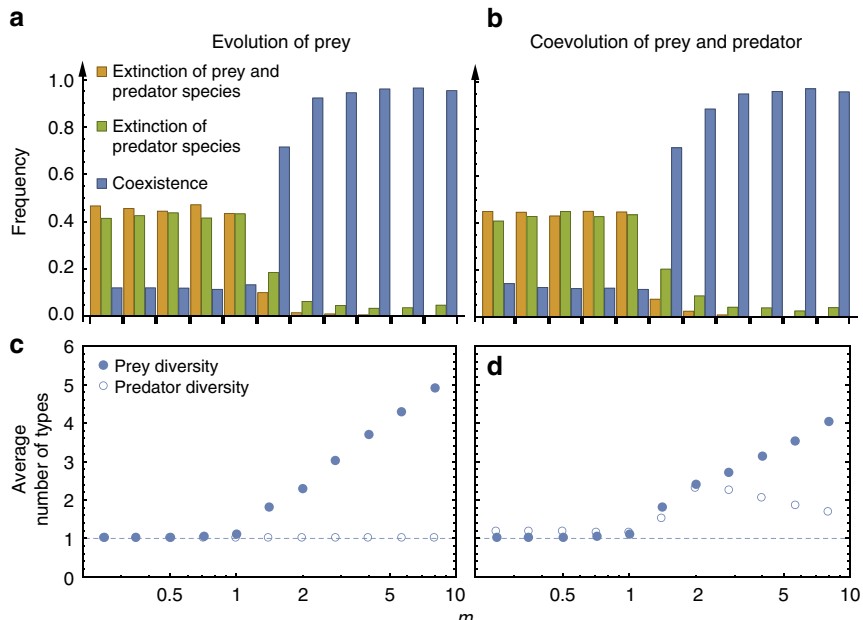

**Fig. 5** Intraspecific diversity with predator–prey coevolution and the evolution of only prey species. Under both processes, the prey and predator species are more likely to coexist when $m > 1$ (**a**, **b**) — the predation function is concave and thus defense is effective and cheap. When the prey and predator species coexist, the average number of prey types increases with $m$ under both processes (**c**, **d**). However, prey species diversity is higher if only the prey species evolves (filled circles in **c**) compared to coevolution (filled circles in **d**). With coevolution, the predator diversity first increases then decreases with $m$ (open circles in **d**). For large $m$, prey defense is cheap and all predator types evolve to a relatively high predation ability, which decreases the predator diversity (parameter set: $b_x = 1.0$, $d_x = 0.1$, $\mu_x = 0.0001$, $r_c = 0.00005$, $d_y = 0.5$, $\mu_y = 0.001$, $p = 0.005$, $k_{max} = 0.3$, $X_1(0) = 1000$, $Y_1(0) = 100$, $g_1 = 1.0$, $k_1 = k_{max}$, averaged over 1000 independent runs and time period equals to 2000 per each run. Note the values for parameter $m$ in the upper panels correspond to the points in the lower panels)

coevolution may result in a change of trade-off shapes, they are beyond the scope of our model. Similar to our experimental results comparing the prey growth-defense trade-offs in naive and coevolved predator populations, the prey growth-defense trade-off curve evolves from a more concave (cheap) to a more convex (costly) shape, when a naive ancestor predator type ($k_l = k_{max}$) improves its predation capability (from Fig. 3a–c). However, different predator types arise from de novo mutations and the $k$ values of the mutant predators can increase or decrease compared to their parents. The trade-off curve may change its shape in different directions being more convex (costly) or concave (cheap) over time.

**Population dynamics and dynamical trade-offs**. In the simplest scenario of a single prey and predator type, we observe a stochastic version of the classical predator-prey cycles with a one-quarter phase shift at the beginning of our simulations. Over time, prey and predator evolve (Fig. 4), leading to changes in the population dynamics depending on the shape of the trade-off. When the initial prey defense is relatively costly (e.g., $m = 3$), we see a shift towards out-of-phase cycles when distinct prey types coexist and a return back to classical cycles when a single prey type dominates (See Supplementary Fig. 4a and a detailed discussion in Supplementary Note 4). When the initial prey defense is effective and cheap (e.g., $m = 10$), we see more often multiple prey types and out-of-phase cycles are prevalent (Supplementary Fig. 4b). Previous models with only prey evolution predict a similar shift in population cycles if distinct prey types can coexist[16, 29].

**Coexistence and diversity**. We next explored the role of the dynamical trade-off on predator and prey diversity. When the initial defense of prey is costly (i.e., $m < 1$ under an ancestor

predator with $k = k_{max}$), it is unlikely for highly defended prey to evolve (Supplementary Fig. 6). In this situation, prey individuals are mostly undefended or have a lower level of defense, which allows the predator species to reach larger population sizes and in turn further increases predation pressure on prey. As a consequence, the predator–prey cycles reach higher peaks and lower minima. This increases the extinction risk of prey and can finally lead to the extinction of both species, which was frequently observed in our simulations. For $m < 1$, we see a higher frequency of the extinction of both species as well as the extinction of only predator species (Fig. 5a, b). On the contrary, when the growth-defense trade-off is initially cheap ($m > 1$), the prey and predator mostly coexist.

Moreover, when both species coexist, the intraspecific diversity differs between the evolutionary and coevolutionary scenario due to the dynamical trade-off (Fig. 5c, d). An initially convex trade-off ($m < 1$) leads to low prey diversity of almost only one prey type with highest growth (see Fig. 5c, d and the invasion analysis in Supplementary Note 5). As defense is costly, even if a prey largely decreases its growth rate, it will not lead to a big improvement of its defense. Thus, it is hard for prey to evolve a different growth rate. When $m$ increases, defense becomes cheaper and multiple prey types with different $g$ values coexist (see Supplementary Figs. 7 and 8, and a discussion of differences among present types in Supplementary Note 6). Thus, the prey diversity increases with an increasing $m$, which holds both for prey evolution and predator-prey coevolution.

However, for a given $m$, the prey diversity is lower with coevolution (Fig. 5c, d). This is because the shape of the trade-off curve in general becomes more convex (costly) under coevolution compared to the evolution scenario with a naive ancestor predator type ($k_1 = k_{max} = 0.3$). On the contrary, if the predator type in the evolution scenario is highly evolved (e.g. $k = 0.06$),

starting with the same predator type coevolution may drive the trade-off curve to be more concave (cheaper) and leads to a higher prey diversity than the evolution of only prey (see Supplementary Fig. 9 and a detailed explanation in Supplementary Note 7).

Regardless whether the populations start with a naive or highly evolved predator, the predator diversity does not monotonically increase with prey diversity (also see the Shannon diversity in Supplementary Note 8). Our results on prey and predator diversity under the coexistence of the two species are independent of realization time. We observe a similar pattern when the model is run for 1000 time steps and for an addition 1000 times steps. The predator diversity stays at intermediate and constant level for $m < 1$, increases with increasing prey diversity for $m > 1$ and reaches a peak at $m \approx 3$, but drops again for $m > 3$. When $m$ is large — effective and cheap defense — all prey types can afford to have a high defense with a low cost of growth, and thus the predator types are constrained to have a high predation ability with a low diversity.

If a prey type with an extremely high defense level arises and spreads in the prey population by chance (Supplementary Fig. 4c), the predation rate of the predator may significantly decrease and leads to a sudden drop of the predator abundance. The adaptation and evolutionary potential of the predator is further constrained by its small population size which results in a shortage of future mutation supply. As demographic stochasticity is higher in small populations, the predator may also go extinct (Supplementary Fig. 4d). Without predation pressure, the prey population has a single fitness peak, which results in a diminishing return in prey growth to the fitness maximum at $g = 1$ (see a further discussion in Supplementary Note 9). This further demonstrates the importance to consider feedbacks between the evolutionary process and demographic fluctuations.

## Discussion

We have hypothesized that trade-offs can be dynamical and their shapes change over time with coevolution of interacting populations. While the growth-defense trade-off in consumer-resource systems has been shown among prey types[30–33], we demonstrate here a dynamical trade-off in a predator–prey system from a coevolution experiment. We further developed a stochastic model with random mutations and demographic fluctuations to explore the impact of such dynamical trade-offs on species coevolution. Both model and experiment show qualitatively the same result in terms of the shape of the trade-off curve. The trade-off curve is concave with cheap defense in prey in the presence of ancestral (non-evolved) predators, but convex with costly defense in the presence of coevolved predators. The results of our coevolutionary model agree with previous theoretical and experimental findings on population dynamics when only the prey evolves, i.e. out-of-phase cycles may arise when distinct prey types coexist.

By comparing scenarios with only prey evolution and predator-prey coevolution, we found a lower prey diversity in coevolving communities compared to the evolution of prey under a naive predator population. This is particularly striking when the initial trade-off curve of the prey is concave. Prey diversity increases with the concavity of the trade-off curve, and the predator diversity increases first, but decreases again when the trade-off curve is extremely concave. When the trade-off curve is extremely concave, all prey types can obtain high defense with little cost. This limits the diversity of predators. Consumer diversity is thus not necessarily positively related to the resource diversity and the role of coevolution for diversification can vary depending on the trait distribution in interacting species over time. This result might help explaining patterns of coevolution and diversity

observed in nature such as coevolutionary hot and cold spots[34]. Our result could also explain the observed asymmetrical evolution between host and parasites in bacteria–phage or algae–virus systems, where the phage or virus could not evolve further to overcome host resistance[24, 35].

The dynamical trade-off curve with coevolution might predict the future evolutionary trajectory of a system. In our case, a concave (cheap) growth-defense trade-off allows large changes in prey traits because prey types with different growth can coexist when defense is cheap to evolve. However, under a convex curve such large changes in traits are likely disadvantageous, as a small change in growth can be detrimental for prey in the presence of predation. Instead, the further evolutionary process under a costly trade-off curve is more likely to be a fine tuning with small changes moving to a fitness peak. Thus, the current shape of a trade-off curve reveals the magnitude of future trait changes.

In summary, our experiment confirms an evolving trade-off curve and our theoretical model shows the potential importance of such dynamical trade-offs in a coevolving predator–prey system. However, we expect a general existence of dynamical trade-offs in other coevolving systems. In addition, our stochastic model presents a simple but generic way to model species interactions with de novo mutation. It captures eco-evolutionary feedback between population size and future mutation supply, and recovers classical population genetics results such as diminishing fitness return — the fitness effects of next fixed beneficial mutations decrease as a population approaches its adaptive peak, which is commonly observed in long-term selection experiments[36–38]. It is straight-forward to extend our model beyond predator–prey systems by changing the concrete definitions of the reactions. Our model provides a general framework to study the evolution of trade-offs and their influences for other species interactions, for example, host–parasite co-evolution and mutualism.

## Methods

**Experimental coevolution of a bacteria-ciliate system.** From a single colony of the bacterium (*Pseudomonas fluorescens* SBW25) and axenic cultures of the ciliate (*Tetrahymena thermophila* 1630/1U (CCAP)), the two species are cultured together in 25 ml glass vials containing 6 ml of the 5% King's B culture medium[39]. We use a serial transfer method and 1% of the culture is transferred into a new vial containing fresh culture medium with a weekly interval. This has been repeated for 6 months before we isolate populations for the measurement. Ciliate control lines (naive ciliates) are kept in an organic medium axenically. After 6 months, bacteria and ciliates are isolated from the selection lines, and ciliates from the control lines. Coevolved ciliates are isolated from cultures where bacteria and ciliates coevolve for ~1000 bacteria and ~550 ciliate generations. For the bacteria samples, 0.5 ml subsample from each vial is frozen with 0.5 ml of 80% glycerol and kept at −80 °C for later analysis (ciliates do not survive freezing under these conditions). Ciliates from the coevolution lines are treated with antibiotics to kill of the bacteria and kept as axenic lines in sterile organic medium. As the antibiotic treatment is only for ~5 generations, we do not expect that there is additional selection acting on these populations in comparison to the control ciliates, which are also treated with antibiotics at a different time point.

After thawing cryopreserved bacterial populations, we grow samples in liquid culture (1% King's B) for 24 h, corresponding to ~10 bacterial generations, so that the subsequently measured phenotypic differences resulted from evolutionary change rather than any induced effects. Growth rates of ancestral and coevolved bacteria are measured in 24-well plates in medium containing M9 salts and King's B (KB) nutrients at 1% concentration (1% KB: 0.2 g/l Peptone number 3 and 0.1 ml/l glycerol). Plates are inoculated under constant shaking at 150 r.p.m., and after 48 h, optical density is measured with a Tecan Infinite spectrophotometer at a wavelength of 600 nm. Naive (ancestral) and coevolved ciliate growth rates are measured on ancestral and coevolved bacteria. Therefore, we add 100 µl of the respective bacteria overnight culture into 2 ml of fresh culture medium, and add 2100 ciliates of the respective origin.

**Modeling prey and predator coevolution.** We implement a minimal representation of the prey type $X_i$ by a growth-defense trade-off. The prey trait is defined by a variable $g_i \in (0, 1)$. A prey with $g_i = 1$ has a maximal growth rate and minimal defense. Similarly, $g_i = 0$ corresponds to a prey type with minimal growth rate and maximal defense. New prey types with different $g_i$ are introduced by de novo

mutations. Without predator, the prey dynamics is captured by birth without (Eq. (3a)) or with mutation (Eq. (3b)), death arising from resource competition (Eq. (3c)), and intrinsic death (Eq. (3d)). Each event occurs with a type-specific rate indicated by the symbols above the arrows,

$$X_i \xrightarrow{b_x g_i (1 - \mu_x)} X_i + X_i, \tag{3a}$$

$$X_i \xrightarrow{b_x g_i \mu_x} X_i + X_j, \tag{3b}$$

$$X_i + X_l \xrightarrow{r_c} X_l, \quad X_i + X_l \xrightarrow{r_c} X_i, \tag{3c}$$

$$X_i \xrightarrow{d_x} 0. \tag{3d}$$

Here, $i, l = 1, 2, ..., n$, denote different prey types and $n$ is the total number of prey types. In our model, $n$ is not fixed. It increases by new prey types (arising from de novo mutation) and decreases by the loss of prey types (extinction). The parameters $b_x$, $d_x$, $r_c$ and $\mu_x$ are kept constant and are identical for all prey types. The reproduction rate of a prey type $X_i$ is its individual growth-defense trade-off parameter $g_i$ multiplied by the common prey birth rate $b_x$. Thus, a larger $g_i$ implies a higher reproduction rate. A new random type $X_j$ occurs with a (small) probability $\mu_x$ during the reproduction of a type $X_i$. This new type is defined by its own growth-defense trade-off parameter $g_j$, which is drawn from a uniform distribution with values between 0 and 1 (see a discussion of normally distributed mutations in Supplementary Note 2). Prey individuals compete for a common resource, and the strength of resource competition is indicated by $r_c$. Note that $X_l$ refers to any prey type currently present in the population, thus here the resource competition of prey within and between types is the same. A homogenous prey population without predation grows logistically in our model. The prey population persists and fluctuates around its carrying capacity if its population size is large enough to avoid extinction due to stochastic effects[28, 40]. A large $r_c$ refers to stronger resource competition and a lower carrying capacity. Furthermore, in the absence of predation pressure, prey types with higher reproduction rates have a fitness advantage. Consequently, in isolation, the prey population evolves towards $g = 1$, where also the equilibrium population size is largest.

Similar to the prey, a predator type $Y_l$ is defined by a trade-off between predation ability and reproduction efficacy $k_l$. The reproduction efficacy is here defined as the ratio of the reproduction rate to the predation rate of predators, thus $k_l < 1$. Mutations occur with a probability $\mu_y$ per reproduction event. Novel predator types have a higher predation ability and lower reproduction efficacy or vice versa. The mutant type $Y_h$ has a reproduction efficacy of $k_h$, which is drawn from a uniform distribution between 0 and $k_{max}$, the upper limit of the reproduction efficacy. Again we show the results under a normal distribution around the parent type (see Normally Distributed Mutations in Supplementary Note 2). Predation without (Eq. 4a) or with reproduction (Eqs. 4b, c)), and death (Eq. 4d) happen with type-specific rates and are given by

$$X_i + Y_l \xrightarrow{(1 - k_l) f(g_i, k_l)} Y_l, \tag{4a}$$

$$X_i + Y_l \xrightarrow{k_l f(g_i, k_l)(1 - \mu_y)} Y_l + Y_l, \tag{4b}$$

$$X_i + Y_l \xrightarrow{k_l f(g_i, k_l) \mu_y} Y_l + Y_h, \tag{4c}$$

$$Y_l \xrightarrow{d_y} 0. \tag{4d}$$

The predation rate, $f(g_i, k_l)$, depends both on the prey defense $g_i$ and the predation ability $k_l$. The more energy the prey spends on its growth $g_i$, the worse it is in defense and escaping predation — $f(g_i, k_l)$ is an increasing function of $g_i$. The more energy the predator spends on its reproduction $k_l$, the worse it is in predation — $f(g_i, k_l)$ is a decreasing function of $k_l$. One simple function that satisfies these two conditions is $f(g_i, k_l) = p \, g_i^{m \frac{k_l}{k_{max}}}$, where $p$ is a general scaling coefficient for the predation rate to the speed of other possible reactions, and $m$ determines the initial shape of the growth-defense trade-off curve for the prey species. We used the same function form (a power function) to fit our experimental data.

**Simulations**. In every realization, we start from the same ancestor prey type (the lowest defense, $g_1 = g_{max} = 1.0$) and predator type (the lowest predation ability, $k_1 = k_{max} = 0.3$) and the same initial population sizes, i.e., 1000 prey individuals and 100 predator individuals. Random mutations arise in both species with rates $\mu_x = 0.0001$ and $\mu_y = 0.001$. Note the important quantities are effective mutation rates, which are $\mu_x$ or $\mu_y$ times the corresponding population sizes of prey or predator. As it is computationally expensive to simulate large populations in individual-based models, we chose the $\mu_x$ and $\mu_y$ values accordingly. During reproduction, a prey can produce a mutant with a different growth rate, which is a uniformly distributed

random number between 0 and $g_{max} = 1.0$. Similarly, a predator can produce a mutant predator type with a different predation capability thus different reproduction efficiency, which is a uniformly distributed random number between 0 and $k_{max} = 0.3$. The evolution of the prey is under a trade-off of growth and defense, the initial shape of which is defined by the parameter $m$. We run 1000 independent realizations under various $m$ (see figure captions) to investigate the impact of different trade-off curves on the coevolutionary dynamics. Please see details for alternative starting conditions, mutant distributions and larger $k_{max}$ values in Supplementary Notes.

**Code availability**. The codes for simulations are available in Github. https://github.com/WeiniHuangBW/DynamicalTradeoffandCoevolution.

**Data availability**. The authors declare that all data supporting the findings of this study are available within the paper and Supplementary Data 1.

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

## Acknowledgements

This work was supported by the German Research Foundation (DFG) BE 4135/3-1 (L.B.), Geoffrey W Lewis Post-Doctoral Grant (B.W.), the Max Planck Society (W.H., A.T.), Cancer Research UK (W.H.), Ministry of Education of China (W.H.), WHRI-ACADEMY (W.H.). We thank J. Cairns for providing the coevolved strains.

## Author contributions

W.H., A.T., B.W. and L.B. developed the model. L.B. and T.H. did the experiments. W.H., B.W., and L.B. analyzed the data. W.H., A.T., B.W. and L.B. wrote the manuscript. All authors read and approved the manuscript.

## Additional information

**Competing interests:** The authors declare no competing financial interests.

