## [Peer Review File · Nature Communications]

Reviewers' comments:

Reviewer #1 (Remarks to the Author):

This study looks at modelling a co-evolutionary predator-prey system, where trade-off shape is itself allowed to evolve, to confirm the results of a biological experiment confirming this behaviour.

The aim of the study is very interesting, and will really help modellers further understand the true behaviour of trade-off functions. For far too long now have more and more modelling studies been published with a lack of understanding on the trade-off itself – something which is vital in determining the behaviour of the model. This article may help make a vital step forward in this area. However, they are potentially missing an opportunity in linking the model and the experiments more. (See point 1 below.)

1) How did the authors choose their baseline parameters for their model? Were they based on the biological experiments, which would be really good if they were, or were they arbitrary values chosen to demonstrate the outcome?

If they were based on the experiments, that is great. However, if the parameters were arbitrarily chosen, then I feel this study could be strengthened somewhat by trying to link the model and experiments further. For example, if they assumed the model was a good approximation for the experimental behaviour, it would be very interesting to see if they could infer what the parameters, mutation rates or trade-off function would be to model the system. There is very little data available on what trade-off shapes, and their functional forms, are actually like, and this study seems like a great opportunity to gain some of this information. It would make a vital contribution to the field. It would also help link the model and experiments better, especially as they are both showing the same results.

Minor points

2) I found it very difficult to fully understand the figures. For example in Figure 1, I am not clear on what the different shapes mean; they say the triangles are the ancestral strains of bacteria, but I do not know the difference between the squares, filled circles and hollow circles. Similarly, on Figure 3, what do the different coloured lines represent? (Also, on line 141, they refer to Fig 2c&d, but there is no Fig 2d?)

3) There needs to be some more clarity when the authors talk about diversity – by which I am assuming they mean the number of co-existing strains of bacteria. Particular, how wide is this diversity in terms of traits (and trait values)? If the differences between the strains (traits) is very small, it may be possible to have a larger number of co-existing strains, compared to when the differences between the strains is larger. It would be more useful to talk about both the number of co-existing strains, and the actual variation between the strains. Also, is the diversity simply due to the length the simulations are run for, and if it were run for longer, would the diversity decrease as strains died out? (It may be that Figs 1&2 in the Supplementary Material begin to answer this point, but again, they are not

clearly explained.)

4) Between line 86 and 87, the authors say they taken mutations to be “drawn randomly from a uniform distribution between 0 and 1”. Would it not be more realistic to take it from a normal distribution, based around where the parent trait is? Would it change the results?

Reviewer #2 (Remarks to the Author):

This study examined how the growth-defense tradeoff can change as the results of coevolution occurring in the predator-prey system. The authors provided the experimental results showing the change of the shape of the tradeoff, and theoretical modelling was used to support the empirical results. Also, the impact of the changing tradeoff on the eco-evolutionary dynamics and the diversity of predator and prey types were explored by the modelling. Overall, their findings and claims are relatively novel and of great interest to researchers in ecology and evolution. I have several questions and concerns that might influence the interpretation of the experimental data and simulation results.

1. Figure 1 shows the experimental results for the measurement of the tradeoff. One ancestral and three coevolved bacterial prey strains were used to measure the growth rate and the defense level against the ancestral or coevolved ciliate predator strains. This experimental design is fine because coevolved bacterial strains share the same ancestor and thus any mutation contributing to the change in growth and defense should be under the same constraint that produces the tradeoff. It would be helpful if the authors explain such the background for why they chose this experimental design, as it took some time for me to understand what was going on in this figure.
2. It is not well explained how the three coevolved bacterial strains were isolated from the coevolving culture. Were they isolated randomly from the single or multiple cultures? How long did they coevolved with ciliates? Why were only three strains isolated though there should be much more diversity in the cultures?, etc. It should be explained enough for the repeatability of the experiments.
3. The experimental procedure should be more explained for the justification for using the specific three coevolved stains in this study, as well. Basically, the two coevolved strains evolved the defense only against the ancestral ciliate, and the one strain evolved defense against both ancestral and coevolved ciliates. Why did only the one strain (shown by the square symbols in Figure 1) evolve so? Was there no strain that broke the constraint so as to have higher defense and higher growth?
4. Also, how were the coevolved ciliates obtained? Apparently, there was no counter-adaptation of ciliates by increasing the predation capacity. Why? It should be related to how the ciliate strains were isolated.
5. Did the authors try to measure reproduction efficacy of ancestral and coevolved ciliates? Was it as assumed in the model? As this tradeoff in ciliate should be important for the later theoretical analysis of eco-evolutionary dynamics and intraspecific diversity, it would be better to show some experimental evidence, if existed.
6. The model assumed the mutation that always decrease the growth (or increased the defense) of prey and decrease the reproduction efficacy (or increase the predation capacity)

of predator. What will happen if mutation occurs to produce g_i being more than 1 and k_l being more than k_{max} ? Do the conclusions of the modelling part hold the same, or what is the rationale behind the boundary of $g_i < 1$ and $k_l < k_{max}$?

7. Also, what will happen if the simulation starts from the small g_i and k_l ? Is there no dependence on initial conditions or bistability? If so, gradual, incremental mutation might produce different results, compared to the global mutation with boundaries used in the present modelling. Did the authors try to compare the results? This question is to see the robustness of the modelling results.

8. It would be helpful if the rationale behind "For any given k_l , the shape of the predation function along various g_i is fixed." is explained, because if the prey evolved to break the constraint making the tradeoff (e.g. better resource acquisition efficiency to have more resources allocated to traits), the assumption would not hold and may influence the conclusions.

9. Figure S1c,d,e,f and S2: it would be informative to show the dominance or relative frequency of different prey and predator types, so that coevolution of prey and predator types might be easily seen in the figures. This is just a suggestion.

Reviewer #3 (Remarks to the Author):

The idea of dynamic trade-offs, i.e., the trade-off structure of a species changing due to the evolution of an interacting species, is both important and interesting. Taking the example the ms focuses on, a species of prey might defend against a non-adapted predator with relatively little effort and thus little cost e.g. to its own fecundity; but if the predator has invested into exploiting this prey more efficiently, then it could be much more costly for the prey to escape. Other exploiter-victim interactions may exhibit similar dynamic trade-offs.

Interesting as the idea is, I was disappointed by the model and its analysis. The model is entirely phenomenological and does not build on the biology behind the change in the trade-off structure. Given the function f defined in line 95 of the ms and the assumption that the predator starts with $k = k_{max}$, it is coded into the assumptions that the trade-off will evolve to be more convex. This should rather emerge as a prediction from a model incorporating the biology of investments and returns.

The model is quite simple and would be amenable to at least some mathematical analysis. Instead, the authors report only stochastic simulations. While simulations can nicely incorporate realistic features, by their very nature they give less insight into why the model behaves as it does. Unfortunately, this weakness is very visible in the present ms, which offers mainly descriptive results. The simulations show that (i) coevolution can lead to either quarter-shift cycles (classical) or out-of-phase cycles (similar to Yoshida et al 2003, who assumed prey evolution only), and (ii) convex trade-offs, produced by coevolution, reduce prey diversity. In the first result, the details leading to the two different outcomes (Fig S1a,b) remain unexplained. The paragraph in lines 112-123 is unclear; if convex trade-offs lead to out-of-phase cycles (line 119), then why does the trade-off evolving to be more convex makes the out-of-phase cycles to be less likely (line 121)?

For (ii), the authors state that convex trade-offs imply less prey diversity. I do not see why this should be the case. Using evolutionary game theory, it is easy to see that a concave trade-off implies a (local) ESS, i.e., concave trade-offs select for a single prey strain. The argument in lines 185-187 is incorrect; with a strongly concave trade-off, there is a wide range of growth rates that all produce similarly high defence, and obviously the highest growth rate is then the best; and also a wide range of defence levels that all produce similarly high growth rates, and then the best defence is the best. This selects for a single prey strain at about the point where the trade-off curve switches from strongly increasing to almost flat.

To quantify diversity, Figure 4 shows the average number of strains as a measure for diversity. This can be seriously misleading: It is not only the number of strains that matters, but also how different they are. A number of strains close to the ESS can linger for a long time, but the coexistence of nearly identical strains does not amount to high diversity. For example, Figure 4d shows ca the same number of predator strains at $m=3$ and $m=10$, but in Figure S1, the predator population appears to be more diverse with $m=3$ than with $m=10$ (the long-lasting predator strains strongly cluster in the latter case but not in the former).

It was confusing to me that Figure 2 plots the inverse of the function f . A convex trade-off means a convex curve in Figure 2 but a concave trade-off function f .

We sincerely thank the reviewers for their time and efforts spent on our work on evolving trade-offs. We considerably extended the manuscript based on the questions and suggestions of the reviewers. Please find a point-by-point response below.

Reply to the reviewers:

Reviewer #1

This study looks at modelling a co-evolutionary predator-prey system, where trade-off shape is itself allowed to evolve, to confirm the results of a biological experiment confirming this behaviour.

The aim of the study is very interesting, and will really help modellers further understand the true behaviour of trade-off functions. For far too long now have more and more modelling studies been published with a lack of understanding on the trade-off itself – something which is vital in determining the behaviour of the model. This article may help make a vital step forward in this area. However, they are potentially missing an opportunity in linking the model and the experiments more. (See point 1 below.)

We thank the reviewer for the positive and instructive comments. We have included the additional aspects suggested by the reviewer.

1) How did the authors choose their baseline parameters for their model? Were they based on the biological experiments, which would be really good if they were, or were they arbitrary values chosen to demonstrate the outcome?

We completely agree with the reviewer that it would be ideal if we could parameterize our model with only experimental measurements. While the absolute values of parameters are difficult to obtain from experiments, we used a minimal set of parameters to model such a coevolution process and chose the parameter combinations with reasonable biological motivations. We added a table to list all our parameters in the end of SI Section 1 and explained the rationale of our parameter values (see SI Lines 33-40).

In our experiments, we didn't observe large death events in the prey population even without nutrition for a certain time period, as it is usually found in unicellular organisms. Thus, we assumed that the

intrinsic death rate d_x is much smaller than the baseline birth rate b_x . We chose the mutation rates of prey and predator, μ_x and μ_y , accordingly to their population sizes. Thus, the two species coevolve on a similar time scale. We chose the other parameters, r_c , p and d_y , under the condition that classical predator-prey cycles are recovered in the simplest control setup of one prey and one predator.

When the resource competition coefficient r_c is too large, the population dynamics are dominated by prey competition instead of the predator-prey interactions and the predator population does not cycle with the prey population. This is not what we aim to study here. When the scaling coefficient of the predation rate p is too large, the predator will consume too fast and the prey will go extinct. When d_y is too large, the predator population will go to extinction.

If they were based on the experiments, that is great. However, if the parameters were arbitrarily chosen, then I feel this study could be strengthened somewhat by trying to link the model and experiments further. For example, if they assumed the model was a good approximation for the experimental behaviour, it would be very interesting to see if they could infer what the parameters, mutation rates or trade-off function would be to model the system. There is very little data available on what trade-off shapes, and their functional forms, are actually like, and this study seems like a great opportunity to gain some of this information. It would make a vital contribution to the field. It would also help link the model and experiments better, especially as they are both showing the same results.

We fully agree with the reviewer that there is little data on the exact functional shape of the trade-off and more experimental information would truly be desirable. We assumed in our model a power function $g^{m \frac{k}{k_{max}}}$ for the predation to capture the growth defense trade-off of the prey. We defined the prey growth g as the base of the power function, the shape of the function is determined by the exponent $\frac{mk}{k_{max}}$, and the predation rate is the outcome of the function. We later used the functional form for the non-linear regression between prey growth and predator growth (a proxy for predation and defense) in our experimental data. This yields a very good fit with R-squared close to the maximum value 1 (also shown in Page 7, Line 93 above Figure 2). In addition, we

see an expected change in the shape of the function from a cheap trade-off with $\frac{mk}{k_{max}} > 1$ (4.296) for the ancestral predator population, to a costly trade-off with $\frac{mk}{k_{max}} < 1$ (0.233 and 0.141) for the two coevolved predator populations. Thus, our experimental data and the model results/assumption show a good fit. We have now added this discussion also in our manuscript (Page 5, Lines 80-83 and Page 9, Lines 127-129).

Unfortunately, to pinpoint all parameters of the model is extremely difficult. The model is too complex to allow a simple inference of, e.g. the mutation rates of the predator and prey. While this is beyond the scope of this study which is to explore the principal effect of dynamical trade-offs in a co-evolutionary setting, we hope to further investigate some of the effects and potentially narrow down some of the interconnected parameters in future work.

Minor points

2) I found it very difficult to fully understand the figures. For example in Figure 1, I am not clear on what the different shapes mean; they say the triangles are the ancestral strains of bacteria, but I do not know the difference between the squares, filled circles and hollow circles.

We apologize for the lack of clarity. The different symbols represent three coevolved bacteria populations. They differ in their growth rates in an increasing order of squares, filled circles and hollows, which correspond to their values in the y-axis. We have explained this in the caption of Figure 2 (previous Figure 1).

We also added a new figure to explain this better. In the new Figure 1, Panel (a) shows the growth rates of the ancestral ciliates when growing on ancestral and the three coevolved bacteria populations. Panels (b-d) show the growth rates of the ancestral and two coevolved ciliate populations when growing on the three coevolved bacteria respectively.

Similarly, on Figure 3, what do the different coloured lines represent?

We thank the reviewer for pointing this out. Different coloured lines refer to prey/predator types with a different g/k value. We have now labeled all types with the corresponding g/k values in our new Figure 4

(previously Figure 3).

(Also, on line 141, they refer to Fig 2c&d, but there is no Fig 2d?)

We apologize for our reference to the wrong figure. We meant Figure 5c&d (previously Figure 4c&d) and have corrected this (Page 12, Line 171).

3) There needs to be some more clarity when the authors talk about diversity – by which I am assuming they mean the number of co-existing strains of bacteria. Particular, how wide is this diversity in terms of traits (and trait values)? If the differences between the strains (traits) is very small, it may be possible to have a larger number of co-existing strains, compared to when the differences between the strains is larger. It would be more useful to talk about both the number of co-existing strains, and the actual variation between the strains.

In our previously submitted manuscript, we recorded the numbers of different types over time and diversity refers to their average value. We fully agree with the reviewer that differences among strains are important. We extended on the measurement of diversity by (1) adding new figures showing the distribution of g/k values to illustrate the difference among types (strains), see SI Figure 5 and 6. (2) We implemented a threshold for the minimal required number of individuals to contribute to diversity, in order to exclude those rare types solely popping up due to randomness. (3) We calculated the Shannon entropy ($-\sum_i p_i \ln p_i$ and p_i is the frequency of type i), where both the number of types and their frequency contribute to the diversity measure, see SI Figure 7.

To briefly summarize the results, we see the same diversity pattern by using average number of prey/predator types and the Shannon index (see SI Section 6). We see a wider distribution of trait values (g/k values) in coevolution compared to evolution of prey (see SI Section 5).

Also, is the diversity simply due to the length the simulations are run for, and if it were run for longer, would the diversity decrease as strains died out? (It may be that Figs 1&2 in the Supplementary Material begin to answer this point, but again, they are not clearly explained.)

Based on the comments of the reviewer, we reran all simulations with a longer time (time =2000 instead of 1000 in the previous version). We also compared the same simulations with measurement at multiple time points. When $m < 1$, we see higher frequencies of extinction of predator species at time=2000 compared to time=1000. As the predator population is smaller than the prey population, it is more likely to go extinct under demographic fluctuations. However, if both species coexist, the diversity patterns are maintained over time.

4) Between line 86 and 87, the authors say they taken mutations to be “drawn randomly from a uniform distribution between 0 and 1”. Would it not be more realistic to take it from a normal distribution, based around where the parent trait is? Would it change the results?

We fully agree with the reviewer that different mutation distributions could be more sensible. Uniformly distributed mutations may lead to larger jumps in trait values. For a normal distribution, traits change in smaller steps (depending on the variance of the distribution). Taking the suggestions of the reviewer, we have now also implemented a normal distribution with large and small variances around the parent trait and compared the results between the uniform and normal distribution (see SI Section 7).

The specific choice of the distribution does not change the general observations of different diversity patterns between evolution and

coevolution. These include

- (1) the prey and predator diversity increase with m when $m > 1$;
- (2) prey diversity is higher in the evolution of prey under a naïve predator type than in the coevolution scenario;
- (3) under coevolution, predator diversity increases first and then decreases again with m , which indicates that the consumer diversity does not necessarily change monotonically with the resource diversity under evolving trade-offs.

However, when the variance for the normal distribution is small, we observe a higher frequency of extinction of predator and prey species compared to the results under a uniform distribution. As species evolve more slowly from their current traits under a normal distribution, it might be more difficult to evolve a different trait to escape extinction.

We have discussed these extensions in the manuscript now and we thank the reviewer again for all constructive and helpful comments.

Reviewer #2 (Remarks to the Author):

This study examined how the growth-defense tradeoff can change as the results of coevolution occurring in the predator-prey system. The authors provided the experimental results showing the change of the shape of the tradeoff, and theoretical modelling was used to support the empirical results. Also, the impact of the changing tradeoff on the eco-evolutionary dynamics and the diversity of predator and prey types were explored by the modelling. Overall, their findings and claims are relatively novel and of great interest to researchers in ecology and evolution. I have several questions and concerns that might influence the interpretation of the experimental data and simulation results.

We thank the review for the encouraging comments and address the questions of the reviewer with a detailed response below.

1. Figure 1 shows the experimental results for the measurement of the tradeoff. One ancestral and three coevolved bacterial prey strains were used to measure the growth rate and the defense level against the ancestral or coevolved ciliate predator strains. This experimental design is fine because coevolved bacterial strains share the same ancestor and

thus any mutation contributing to the change in growth and defense should be under the same constraint that produces the tradeoff. It would be helpful if the authors explain such the background for why they chose this experimental design, as it took some time for me to understand what was going on in this figure.

We apologize that we did not explain our motivation and the details of the experiment better. We now added more information to the manuscript (Page 5, Lines 60-83) and added a new figure (Figure 1) to show the differences in bacteria growth and ciliate growth (=defense) clearer.

2. It is not well explained how the three coevolved bacterial strains were isolated from the coevolving culture. Were they isolated randomly from the single or multiple cultures? How long did they coevolved with ciliates? Why were only three strains isolated though there should be much more diversity in the cultures?, etc. It should be explained enough for the repeatability of the experiments.

The bacteria populations were isolated after ~1000 generations of coevolution with the ciliates at the same time when we isolated the ciliates. Bacteria populations were isolated from three individual selection lines and two out of the three bacteria populations came from the same selection line as the ciliates. We did not isolate individual strains but populations, and all measurements are on the population level. We clarified this through out the text. The isolation procedure is explained in the Methods section (Page 17, Lines 249-257).

3. The experimental procedure should be more explained for the justification for using the specific three coevolved stains in this study, as well. Basically, the two coevolved strains evolved the defense only against the ancestral ciliate, and the one strain evolved defense against both ancestral and coevolved ciliates. Why did only the one strain (shown by the square symbols in Figure 1) evolve so? Was there no strain that broke the constraint so as to have higher defense and higher growth?

The reviewer is correct that we did not explain and discuss the results detailed enough. The pattern the reviewer describes is what we would expect with predator-prey coevolution. To illustrate this better, we have

added a new Figure 1 to show this in more detail.

- when the bacteria evolves a defense, the ciliate growth rate should be lower compared to the ciliate growth rate when the ciliate feeds on ancestral bacteria. This is what we see for the all three coevolved bacteria populations (Figure 1a).
- when the ciliate coevolves, the coevolved ciliates should have a higher growth rate compared to the ancestral ciliates at a given coevolved bacteria population. This is what we see in our data (Figure 1c-d).

We present in the manuscript the results for all isolated bacteria and ciliate populations. We didn't see a break of the trade-off constraints with high defense and high growth.

4. Also, how were the coevolved ciliates obtained? Apparently, there was no counter-adaptation of ciliates by increasing the predation capacity. Why? It should be related to how the ciliate strains were isolated.

As described above and now in the Methods section, coevolved ciliates were isolated from the selection lines after ~550 generations and treated with antibiotics to remove bacteria. As the antibiotic treatment was only for ~5 ciliate generations, we do not expect that there was additional selection acting on these populations in comparison to the control ciliates. Isolated ciliate populations were kept in bacteria-free organic medium. Furthermore ancestral ciliates underwent the same antibiotics treatment but at a different time point. We added this information to the Methods section.

The coevolved ciliates (Figure 1c-d blue and green bars) showed a higher growth rate when feeding on all three coevolved prey compared to the ancestral ciliate (Figure 1c-d black bars), which we interpret as a counter-adaptation to evolving defense levels of the prey.

5. Did the authors try to measure reproduction efficacy of ancestral and coevolved ciliates? Was it as assumed in the model? As this tradeoff in ciliate should be important for the later theoretical analysis of eco-evolutionary dynamics and intraspecific diversity, it would be better to show some experimental evidence, if existed.

We agree with the reviewer that it would be ideal if we could measure

the trade-off in ciliates as well. Unfortunately, our current measurement of predator growth does not allow us to separate reproduction and predation efficacy. Thus, it is an assumption of our model at this moment. We state this now clearly in our manuscript (Page 9, Lines 129-130).

6. The model assumed the mutation that always decrease the growth (or increased the defense) of prey and decrease the reproduction efficacy (or increase the predation capacity) of predator. What will happen if mutation occurs to produce g_i being more than 1 and k_l being more than k_{max} ? Do the conclusions of the modelling part hold the same, or what is the rationale behind the boundary of $g_i < 1$ and $k_l < k_{max}$?

We apologize that we did not explain the motivation of our boundaries clear enough. We started the experiments with the naïve predator and prey type. We agree with the reviewer that the first mutations will always increase defense/ predation capability. However, this is not the case over time, as mutations are random and new types with either higher or lower k/g values compared with the parent type are introduced. We have now explored different initial condition (SI section 9), where the prey defense and predation capability can decrease initially (please also see our reply to the next point).

We set up upper boundaries for g and k , because these two quantities are parts of the reaction rates (Eqs.1&2 in SI), which should be positive but finite values. The maximum value for g in principle can be any value. Here, we scaled the system between 0 and 1 (Page 8, Lines 107-109). k is the ratio of predator reproduction in the total predation (Eqs.2a&2b in SI). Because the predator needs to consume to reproduce offspring, k_{max} has to be less than 1. For an extreme but not realistic case, $k_{max}=1$, every time a predator consumes it reproduces an offspring. Thus, we chose a small $k_{max}=0.3$ rather than large k_{max} value in our simulations (Page 8, Lines 114-117). We now show the choice of k_{max} will not change the deterministic dynamics of the prey and predator (Figure SI 1).

However, we have extended our stochastic simulations to a larger $k_{max}=0.6$ (see SI Section 8). As the predator can produce more offspring by consuming the same amount of prey under larger k_{max} values, we observe a higher extinction of prey and thus a higher extinction of both

species (Figure SI 10) compared to $k_{\max}=0.3$ (Figure 5). Interestingly, our diversity patterns are consistent under small and large k_{\max} when prey and predator coexist (see more details in SI Section 8).

7. Also, what will happen if the simulation starts from the small g_i and k_i ? Is there no dependence on initial conditions or bistability? If so, gradual, incremental mutation might produce different results, compared to the global mutation with boundaries used in the present modelling. Did the authors try to compare the results? This question is to see the robustness of the modelling results.

We fully agree with the reviewers and have done considerable extensions of our model by running additional simulations and adding new analytical work. As we explained above, we started with a maximum g and k because we were thinking of the most natural starting point for a co-evolving system, where prey and predator have not been exposed to each other. We now also simulated different initial conditions with small g and k values (e.g. $g=0.2$, $k=0.06$, Figure SI 11 and SI Section 9) as well as gradually changing mutations drawn from a normal distribution around parent types (SI Section 7, Figure SI 8 and 9, please also see details on our answer to point 4 of the first reviewer).

In general, our results on diversity patterns and differences among evolution and coevolution scenarios are the same, although different initial conditions and distributions to draw mutations do change the extinction probabilities and the diversity values quantitatively.

Starting with small g and k values (e.g. $g=0.2$, $k=0.06$ for $g_{\max}=1.0$, $k_{\max}=0.3$), we observe a high frequency of extinction of predators for all m (see Figure SI 11). This is because

- (1) the predator cannot eat the prey as they are highly defended (small g , also can be seen in Figure SI 4d: extinction of predator) and
- (2) even if the predator can eat, it spends too little energy on reproduction (small k). Thus, it cannot increase its abundance sufficiently fast.

However, when prey and predator coexist, we observe similar diversity pattern as before under coevolution.

8. It would be helpful if the rationale behind “For any given k_l , the shape of the predation function along various g_i is fixed.” is explained, because if the prey evolved to break the constraint making the tradeoff (e.g. better resource acquisition efficiency to have more resources allocated to traits), the assumption would not hold and may influence the conclusions.

We agree with the reviewer that the existence of a trade-off is a critical assumption in our model. We intend to explore the population dynamics if a trade-off evolves according to both prey (g values) and predator (k values) under coevolution. As we would like to entangle the contribution of evolution and coevolution, we assume the simplest scenario that the shape of the trade-off (function) does not change if only one species evolves.

We agree there might be more complex scenarios that a change or breakdown of trade-offs can occur without coevolution. However, this is beyond the scope of our model. It would require additional detailed assumptions of e.g. how this growth-defense trade-off function will change under the evolution of prey, for which we have no solid experimental basis. Thus, we used the simplest assumption to first explore the evolving trade-offs due to coevolution and stated this in our manuscript (Page 9, Lines 138-140). However, we fully agree more complex scenarios and mechanism causing changing trade-offs should be explored further and followed up on.

9. Figure S1c,d,e,f and S2: it would be informative to show the dominance or relative frequency of different prey and predator types, so that coevolution of prey and predator types might be easily seen in the figures. This is just a suggestion.

Agreed. We now have new Figures where we show the relative frequencies of prey and predator types by gray areas around the g/k values, see Figure SI 3 and 4.

We thank the reviewer again for all helpful comments.

Reviewer #3 (Remarks to the Author):

The idea of dynamic trade-offs, i.e., the trade-off structure of a species

changing due to the evolution of an interacting species, is both important and interesting. Taking the example the ms focuses on, a species of prey might defend against a non-adapted predator with relatively little effort and thus little cost e.g. to its own fecundity; but if the predator has invested into exploiting this prey more efficiently, then it could be much more costly for the prey to escape. Other exploiter-victim interactions may exhibit similar dynamic trade-offs.

We thank the reviewer for the positive comments on the fundamental idea of our model.

Interesting as the idea is, I was disappointed by the model and its analysis. The model is entirely phenomenological and does not build on the biology behind the change in the trade-off structure. Given the function f defined in line 95 of the ms and the assumption that the predator starts with $k=k_{max}$, it is coded into the assumptions that the trade-off will evolve to be more convex. This should rather emerge as a prediction from a model incorporating the biology of investments and returns.

We understand the concern of the reviewer. In our experiments, we are comparing the trade-offs of preys in naïve predator populations to coevolved predator populations, which is the same order of our Figure 3 from left (a) to right (c) panels.

Our experiments confirmed our hypothesis of dynamic tradeoffs, and motivated us to chose a starting condition with highest g and k . Starting with highest g and k in simulations, the trade-off does evolve to become on average more convex (costly). However, in the co-evolutionary scenario, the trade-off does not necessary only evolve to become more convex (costly) over time. Instead, it depends on which predator is present in the system. To strengthen our conclusions, we now also simulated different initial conditions (see SI Section 9), where the population starts with evolved prey and predator with higher defense and predation capabilities. Our conclusions hold for different patterns of diversity among evolution and coevolution.

The model is quite simple and would be amenable to at least some mathematical analysis. Instead, the authors report only stochastic simulations. While simulations can nicely incorporate realistic features,

by their very nature they give less insight into why the model behaves as it does. Unfortunately, this weakness is very visible in the present ms, which offers mainly descriptive results.

We thank the reviewer for the suggestions. We would like to incorporate the impact of demographic fluctuations including the possibility of extinction in finite populations, which naturally calls for stochastic simulations. We have now added a full section of invasion analysis on prey and predator species based on adaptive dynamics (see SI Section 2, Page 5-8 and Figure SI 1) and discussed the analytical results in the context of the diversity patterns observed in our simulations. This is of course a description that covers only part of our computational model, but it leads to additional insight in terms of expected invasions. More specifically, when $m < 1$ we expect that both g and k will evolve to their maximum values, which is consistent between our invasion analysis and simulations. When $m > 1$, g and k values evolve to intermediate values.

The simulations show that (i) coevolution can lead to either quarter-shift cycles (classical) or out-of-phase cycles (similar to Yoshida et al 2003, who assumed prey evolution only), and (ii) convex trade-offs, produced by coevolution, reduce prey diversity. In the first result, the details leading to the two different outcomes (Fig S1a,b) remain unexplained.

Yoshida et al 2003 showed the existence of out-of-phase cycles under prey evolution. Their study and subsequent studies (e.g. Becks et al. 2010 Ecology Letters) showed that out-of-phase cycles occur when distinct prey types exist in the same population. Here we report a similar observation. We find out-of-phase cycles are more prevalent under more concave (cheap) trade-off curves where multiple prey types with different traits can coexist. Thus, we think the observation of out-of-phase cycles might be independent of an evolutionary or co-evolutionary process. We have revised our discussion on out-of-phase cycles in our manuscript (Page 11, Lines 152-157; Page 13, Lines 208-210) and SI Section 3.

The paragraph in lines 112-123 is unclear; if convex trade-offs lead to out-of-phase cycles (line 119), then why does the trade-off evolving to be more convex makes the out-of-phase cycles to be less likely (line 121)?

We thank the reviewer to point out a mistake we had in our previous version in line 119. We now corrected this to “Previous models with only prey evolution predict a similar shift in population cycles if distinct prey types can coexist ” (Page 11, Lines 156-157). In our model when the trade-off is costly (convex), a single prey strain is selected. While out-of-phase cycles require the coexistence of prey types with different growth and defense, we do not observe out-of-phase when defense is very costly.

In addition, we recognized our definition of concave and convex tradeoffs should be clearer to avoid confusion. We mean concave and convex trade-off by the shape of the curve in Figures 2 and 3, not the function itself (please also see our reply to the last comment of the reviewer).

Although different definitions of concave and convex trade-offs can exist in different research and therefore might lead to confusion, the definition of costly and cheap trade-offs is unique. A cheap trade-off means that a small decrease of growth can lead to a large increase of defense, which refers to concave trade-offs in our model. Thus, we now specifically use cheap and costly when we refer trade-offs to avoid misunderstandings.

For (ii), the authors state that convex trade-offs imply less prey diversity. I do not see why this should be the case. Using evolutionary game theory, it is easy to see that a concave trade-off implies a (local) ESS, i.e., concave trade-offs select for a single prey strain.

We apologize again that our definition of concave and convex may have lead to a misinterpretation. A concave trade-off means defense is cheap to evolve. A convex trade-off means defense is costly to evolve. Consistent with previous models, our analysis shows a single strain is selected under a costly trade-off curve (SI section 2).

The argument in lines 185-187 is incorrect; with a strongly concave trade-off, there is a wide range of growth rates that all produce similarly high defence, and obviously the highest growth rate is then the best; and also a wide range of defence levels that all produce similarly high growth rates, and then the best defence is the best. This selects for a single prey strain at about the point where the trade-off curve switches

from strongly increasing to almost flat.

We agree with the reviewer that we should not use the word “similarly” here. We meant under a cheap tradeoff curve, many prey types can have relatively high defense and thus can survive under the presence of predators. A balance between growth and defense is easier to reach and multiple prey types coexist. We now changed our statement to “in our case, a concave (cheap) growth-defense trade-off allows large changes in prey traits, because prey types with different growth can coexist when defense is cheap to evolve” (Page 15, Lines 225-227).

To quantify diversity, Figure 4 shows the average number of strains as a measure for diversity. This can be seriously misleading: It is not only the number of strains that matters, but also how different they are. A number of strains close to the ESS can linger for a long time, but the coexistence of nearly identical strains does not amount to high diversity. For example, Figure 4d shows ca the same number of predator strains at $m=3$ and $m=10$, but in Figure S1, the predator population appears to be more diverse with $m=3$ than with $m=10$ (the long-lasting predator strains strongly cluster in the latter case but not in the former).

We agree with the reviewer that only measuring the average number of types is not enough. Thus, we extended our measurement to include the Shannon diversity where frequencies of types are considered (Figure SI 7). We also added new figures that show the distributions of g/k values (Figure SI 5 and 6). The diversity patterns measured by Shannon entropy are consistent with those on the average number of types. We observe a larger difference of prey types under coevolution compared to evolution of preys under the same parameters.

In Figure 5d (previous Figure 4d), we show lower number of predator types for $m=8$ compared to $m=3$. This is a key conclusion of our model: The predator diversity increases first and decreases later when m is very large. The diversity of predators does not monotonically increase with prey diversity. This decreasing predator diversity under large m consists with Figure SI 3 (Figure S1 in our previous version) as observed by the reviewer.

It was confusing to me that Figure 2 plots the inverse of the function f . A convex trade-off means a convex curve in Figure 2 but a concave trade-

off function f .

We apologize again for this confusion. We mean concave and convex by the shape of the curve in Figs. 2 and 3, not to the function itself. We plot the inverse of the predation function, because we would like to be consistent with how trade-off curves were plotted in previous work with the defense shown in the x-axis (e.g. Jones, PRSB, 2009). The defense of the prey is traditionally measured by the growth of the predator as in our experiment Figure 1. As the growth of the predator is the inverse of the defense of the prey (as well as the predation function), we plot our Figure 3 (former Fig. 2) in this format. We thank the reviewer for pointing out our imprecisions and potential source of confusion, and we have specifically stated this in our manuscript (Page 9, Lines 131-134) to avoid misunderstanding.

We thank all reviewers again for all the constructive suggestions. We benefited a lot from these comments, which helped us to improve the quality and presentation of our results.

REVIEWERS' COMMENTS:

Reviewer #1 (Remarks to the Author):

I am happy with the changes the authors have made to the manuscript in response to my comments - in fact I am impressed with the amount of extra work they have carried out for them.

My only comments would be that they have added some very useful comments in their response to me, that have not been included in the manuscript itself. A couple of examples are:

1) In the response to point 1, they said they "didn't observe large deaths in the prey population, even without nutrition for a certain time period... we assumed that the intrinsic death rate dx is much smaller than the birth rate bx ". However they didn't include this in the manuscript, which will likely be of interest to readers.

2) In the response to point 3, they ran the simulations for time=2000 (instead of time=1000) to show the diversity was unlikely to be due to the transient dynamics of the model. Again I found this very interesting, and so may other readers. So a simple comment in the main manuscript such as "(These results were seen to hold when the model was run for an addition 1000 times steps.)" would be useful.

Otherwise, I am happy with the manuscript.

Reviewer #2 (Remarks to the Author):

Thank you for clarifying and responding to my questions and comments. I am satisfied with all the revision the authors made.

We sincerely thank the reviewers for their positive comments on the revision of our manuscript. Please find a point-by-point response below.

Reply to the reviewers:

Reviewer #1 (Remarks to the Author):

I am happy with the changes the authors have made to the manuscript in response to my comments - in fact I am impressed with the amount of extra work they have carried out for them.

We thank the reviewer for the positive comments.

My only comments would be that they have added some very useful comments in their response to me, that have not been included in the manuscript itself. A couple of examples are:

1) In the response to point 1, they said they "didn't observe large deaths in the prey population, even without nutrition for a certain time period... we assumed that the intrinsic death rate dx is much smaller than the birth rate bx ". However they didn't include this in the manuscript, which will likely be of interest to readers.

Agreed. We now added this in the section of Rationale of Parameter Choice (Line 4-7) in Supplementary Note 1.

2) In the response to point 3, they ran the simulations for time=2000 (instead of time=1000) to show the diversity was unlikely to be due to the transient dynamics of the model. Again I found this very interesting, and so may other readers. So a simple comment in the main manuscript such as "(These results were seen to hold when the model was run for an addition 1000 times steps.)" would be useful.

We now added this in the Results of the main text (Line 185-187) as follow. "Our results on prey and predator diversity under the coexistence of the two species are independent of realization time. We observe a similar pattern when the model is run for 1000 time step and for an addition 1000 times steps."

Otherwise, I am happy with the manuscript.

We thank the review again for all instructive comments and support on our manuscript.

Reviewer #2 (Remarks to the Author):

Thank you for clarifying and responding to my questions and comments.

I am satisfied with all the revision the authors made.

We are glad that the review is satisfied with our revision. We thank the review again for all efforts spent on our manuscript.